# Integrated Analysis of Cortex Single-Cell Transcriptome and Serum Proteome Reveals the Novel Biomarkers in Alzheimer’s Disease

**DOI:** 10.3390/brainsci12081022

**Published:** 2022-08-01

**Authors:** Qing-Shan Yu, Wan-Qing Feng, Lan-Lan Shi, Rui-Ze Niu, Jia Liu

**Affiliations:** Laboratory Zoology Department, Kunming Medical University, Kunming 650500, China; 20200265@kmmu.edu.cn (Q.-S.Y.); chi08010285@163.com (W.-Q.F.); huangshilanlan@163.com (L.-L.S.)

**Keywords:** Alzheimer’s disease, single-nuclei transcriptome sequencing, biomarkers, cortex, serum, proteomics

## Abstract

Blood-based proteomic analysis is a routine practice for detecting the biomarkers of human disease. The results obtained from blood alone cannot fully reflect the alterations of nerve cells, including neurons and glia cells, in Alzheimer’s disease (AD) brains. Therefore, the present study aimed to investigate novel potential AD biomarker candidates, through an integrated multi-omics approach in AD. We propose a comprehensive strategy to identify high-confidence candidate biomarkers by integrating multi-omics data from AD, including single-nuclei RNA sequencing (snRNA-seq) datasets of the prefrontal and entorhinal cortices, as wells as serum proteomic datasets. We first quantified a total of 124,658 nuclei, 8 cell types, and 3701 differentially expressed genes (DEGs) from snRNA-seq dataset of 30 human cortices, as well as 1291 differentially expressed proteins (DEPs) from serum proteomic dataset of 11 individuals. Then, ten DEGs/DEPs (NEBL, CHSY3, STMN2, MARCKS, VIM, FGD4, EPB41L2, PLEKHG1, PTPRZ1, and PPP1R14A) were identified by integration analysis of snRNA-seq and proteomics data. Finally, four novel candidate biomarkers (NEBL, EPB41L2, FGD4, and MARCKS) for AD further stood out, according to bioinformatics analysis, and they were verified by enzyme-linked immunosorbent assay (ELISA) verification. These candidate biomarkers are related to the regulation process of the actin cytoskeleton, which is involved in the regulation of synaptic loss in the AD brain tissue. Collectively, this study identified novel cell type-related biomarkers for AD by integrating multi-omics datasets from brains and serum. Our findings provided new targets for the clinical treatment and prognosis of AD.

## 1. Introduction

Alzheimer’s disease (AD) is a progressive neurodegenerative disease characterized by senile amyloid plaques and tau fibrillary tangles, representing the main cause of dementia [1,2,3]. Currently, the incidence of dementia is estimated to have exceeded 45 million all around the world. With the development of science and technology, improved living conditions, and an increase in life expectancy, the number of people with dementia is expected to triple by 2050 [4]. The clinical symptoms of AD individuals often start with mild memory loss and eventually evolve into severe impairment of extensive executive and cognitive functions [5]. Symptoms such as cognitive dysfunction caused by AD not only have a huge impact on the daily life of patients themselves, but also cause a heavy burden on the patients’ family and even the whole of society [6]. Patients with AD have traditionally been diagnosed, referring to symptoms and behavioral tests, and confirmed by post-mortem brain pathology [7]. However, there is still no accurate and effective diagnosis method. At the same time, the pathological mechanism of AD remains obscure, and seldom effective therapy has been found [8]. Therefore, it is urgent to screen out biomarkers that can systematically reflect the pathogenesis of AD, thus intervening as early as possible for patients.

Single-cell RNA sequencing (scRNA-seq) is the amplification and sequencing of the transcriptome at the single-cell level [9], which provides an effective method for studying the cellular heterogeneity of the brain and illuminates the complex mechanisms of the normal physiological or pathological development process [10]. At present, scRNA-seq is widely used in the research of neurological diseases, such as AD [11], brain aging [12], and glioma [13]. At the same time, the cellular and biochemical components of blood play a central role in human physiology, and their dynamic levels are thought to correlate with the individual’s health and disease states [14]. Human serum contains a variety of proteins secreted from cells and tissues to achieve normal physiological functions, along with proteins from damaged cells and tissues under disease conditions. Proteomic analysis of serum can further discover important markers, thus revealing the occurrence and development of diseases and providing crucial guidance on the diagnosis and treatment of the disease [15,16].

In this research, we integrated and analyzed snRNA-seq datasets of the prefrontal and entorhinal cortex. The snRNA-seq datasets were selected after careful screening, mainly considering the completeness of donor information (including the age, gender, AD stage, pathological grading, etc., of AD patients) and sample quality (including the number of cells, nuclear genes, and mitochondrial genes). A total of 30 samples were selected, including 18 prefrontal cortex samples (8 from AD patients and 10 from normal elderly) and 12 entorhinal cortex samples (6 from AD patients and 6 from normal elderly), from 3 different datasets (GSE141552, GSE157827, and GSE138852) [17,18,19]. A proteomic dataset of serum (PXD011482) was also included [20] to reveal new key biomarkers for AD. This study unveiled the potency of new targets for the diagnosis, treatment, and prognosis of AD.

## 2. Materials and Methods

### 2.1. Datasets

All snRNA-seq data were obtained from Gene Expression Omnibus (GEO) (https://www.ncbi.nlm.nih.gov/geo/ (accessed on 21 December, 2021)): GSE141552 (containing 4 prefrontal cortex samples from 4 controls), GSE157827 (containing 14 prefrontal cortex samples from 6 controls and 8 AD patients), and GSE138852 (containing 12 entorhinal cortex samples from 6 controls and 6 AD patients). In GSE141552, data from alcoholics were excluded because prolonged alcohol abuse caused individual mental loss, poor attention, and memory loss, which did not meet the inclusion criteria for the AD study. In GSE157827, compared to other samples, 3 controls and 3 AD were discarded, as a result of small number of cells and genes. In the GSE138852 dataset, there was a total of 16 samples (containing 8 controls and 8 AD patients). However, 2 AD and 2 controls were discarded because of abnormal high neuronal enrichment. The first two datasets were obtained using the 10× Genomics platform and NovaSeq 6000 sequencing platform, and the additional datasets using the 10× Genomics and NextSeq 500 sequencing platforms [17,18,19]. The proteomic data was obtained from the Proteome Xchange Consortium: PXD011482 (containing 6 AD and 5 controls).

### 2.2. Patients Informantion and Inclusion Criteria

Twenty healthy elderly people, in the age range of 60–80 years, were recruited at Yunnan Provincial Psychiatric Hospital. All of them met the following inclusion criteria: diagnosed with mild or severe AD; Mini-Mental State Examination (MMSE) score > 19; aged 60–90 years; anti-inflammatory dementia or mood stabilization dosing stabilize medication. They were all informed about the purpose of the study and signed an informed consent form.

### 2.3. snRNA-seq Analysis

#### 2.3.1. Preprocessing, Quality Control, and Data Integration

The gene barcode matrices for each sample were imported into R software using the Read 10X function in the Seurat R package [21]. The Seurat object, corresponding to each sample, was created using the CreateSeuratObject function, with the input gene barcode matrix provided as the raw data. The datasets were integrated using the method of Stuart et al. [22]. Data quality was controlled prior to integration. The number of genes per sample, unique molecular identity counts, and percentage of mitochondrial genes were controlled. To exclude potential dead cells and cell debris from the dataset, we filtered out nuclei with ≤200 genes, ≥2500 unique molecular identifiers, or ≥5% mitochondrial genes. In total, 124,658 high-quality nuclei were obtained for subsequent analyses. For the integration analysis, the highly variable features of each sample were identified using the FindVariableFeatures function. The parameter is selection.method = vst, nfeatures = 2000. To integrate all samples, the features of the samples were anchored using the FindIntegrationAnchors function, with the parameter dims = 1:20. All samples were integrated using the IntegrateData function, with the parameter of dims = 1:20.

#### 2.3.2. Data Dimension Reduction and Clustering Analysis

Subsequently, we scaled the expression matrix and performed a linear dimension reduction using the RunPCA function with the parameter npcs = 50. The *p*-value distribution of each major component was visualized using the JackStrawPlot function and selected to perform graph-based clustering using the first 30 principal components. We performed K-nearest neighbor (KNN) clustering using the FindClusters function with the parameter resolution = 1 and UMAP clustering using the RunUMAP function with the parameter dims = 1:30, which initially yielded 31 cell clusters. We identified the DEGs in each cell cluster by the Wilcoxon rank sum test using the FindAllMarkers function with the parameters logfc.threshold = 0.25 and test.use = wilcox. We then assigned a cell type identity to each cell cluster, according to the expression of known cell type markers, and identified additional cell type-specific marker genes by the Wilcoxon rank sum test using the FindAllMarkers function with the parameters logfc.threshold = 0.25 and test.use = wilcox. For cell type markers, the level of statistical significance was set at an adjusted *p*-value < 0.1.

#### 2.3.3. Examination of Cell Type-Specific Transcriptomic Changes

To examine the cell type-specific transcriptomic changes in AD, we compared the transcriptome profiles of individual cell types among AD and control samples by the Wilcoxon rank sum test using the FindMarkers function with the parameters logfc.threshold = 0.25 and test.use = wilcox. The level of statistical significance for cell type-specific transcriptomic changes was set at an adjusted *p* < 0.1 and log2 fold change ≥0.25 or ≤−0.25.

### 2.4. Gene Ontology (GO) and Kyoto Encyclopedia of Genes and Genomes (KEGG) Signaling Pathway Enrichment Analysis

In this study, DEGs were all imported for analyzing GO terms in the GO database (http://www.geneontology.org (accessed on 10 February, 2022)), and the number of genes for each term was calculated. Pathway-based analysis was used to characterize the biological functions of the DEGs. Pathway enrichment analysis identified significant signal transduction pathways in the KEGG database (http://www.genome.jp/kegg/ (accessed on 15 February 2022)). In GO and KEGG enrichment analysis, R software version 3.8.1 (http://www.r-project.org (accessed on 22 February 2022)) and multiple R packages, such as clusterProfiler, org.Hs.eg.db, enrichplot, and ggplot2, were used to generate the bars, bubble maps, and signaling pathway maps.

### 2.5. PPI Network Analysis

The STRING database (https://www.string-db.org/ (accessed on 25 February 2022)) was used for DEG-associated protein interaction analysis and production of PPI network. Cytoscape 3.8.0 was used (https://cytoscape.org/ (accessed on 25 February 2022)) to construct the cell differential expression network.

### 2.6. ELISA Validation

Blood samples of 20 AD individuals and 20 healthy elderly controls were collected (Appendix A). After coagulation and centrifugation, serum was collected and stored in aliquots of Eppendorf tubes at −80 °C until use. We used the NEBL ELISA (MM-60704h2, MEIMIAN, CN), EPB41L2 ELISA (MM-60714h2, MEIMIAN, CN), the FGD4 ELISA kit (MM-60715h2, MEIMIAN, CN), and MARCKS ELISA (MM-60720h2, MEIMIAN, CN) kits to detect the levels of four proteins (NEBL, EPB41L2, FGD4, and MARCKS) in serum samples. As previously described [23], briefly, the microtiter plates were coated with purified antibodies. Then, blank, standard, and sample wells were set. We added 25 μL sample dilution and 25 μL pending sample to the sample wells. A known concentration of 50 μL of the standard was added to the standard wells. Nothing was added to the blank well. Then, 100 μL HRP-conjugated reagent was added to each well, except for the blank wells, and then incubated in a 37 °C incubator for 60 min after gentle mixing. After rinsing, 50 μL substrate A and 50 μL substrate B were added to each well and gently mixed. After 15 min of incubation in the dark at 37 °C, the reaction was stopped, with 50 μL of the stop solution. The absorbance (optical density (OD) value) was measured for each sample within 15 min by using a spectrophotometer (Thermo Fisher Scientific, Vantaa, Finland) at a wavelength of 450 nm. Finally, the linear regression equation of the standard curve was calculated using the concentration and OD values of the standard product; then, the concentration of each protein in the serum was calculated.

### 2.7. Statitical Analysis and Data Visualization

Differences between AD and controls were analysed using Student’s *t*-test (SPSS v26.0, IBM, USA). *p* < 0.05 was considered statistically significant. We visualized the data using Seurat’s DoHeatmap, DotPlot function, TBtools, and Cytoscape (version 3.8.0), where appropriate.

## 3. Results

### 3.1. Integrative Analysis of snRNA-seq Data from Prefrontal and Entorhinal Cortex

To investigate transcriptional and cellular differences in the prefrontal and entorhinal cortex of AD individuals, we integrated three snRNA-seq datasets using the method of Stuart et al. [22]. The three datasets contained snRNA-seq sequencing data from 8 AD patient prefrontal cortex (PFC_AD, age 74.9 ± 12.0, *n* = 8) (Table 1, Appendix A) and 10 age-matched normal human prefrontal cortex (PFC_Ctl, age 77.4 ± 13.9, *n* = 10) (Table 1, Appendix A), the entorhinal cortex of 6 AD patients (EC_AD, age 78.9 ± 8.5, *n* = 6) (Table 1, Appendix A), and the entorhinal cortex of 6 age-matched normal subjects (EC_Ctl, age 76.4 ± 6.0, *n* = 6) (Table 1, Appendix A). After standardized integration and quality control, a total of 124,658 nuclei were obtained for subsequent downstream processing (PFC_Ctl: 55717, PFC_AD: 55845, EC_Ctl: 6533, EC_AD: 6563).

### 3.2. Cell-Specific Transcriptional Profiles of the Human Brain Cortex

To construct taxonomic maps of cell populations, we integrated 30 subject-derived snRNA-seq data and performed PCA and UMAP cluster analysis. This analysis yielded 31 unique clusters (c0–c30) (Figure 1B). By comparing the DEGs of all cell types, we know the specific genes expressed by each cell type. Subsequently, we could identify the cell types of the 31 cell clusters based on their respective transcriptome expression and previously reported cell type markers. They were finally classified into the following eight cell types: astrocytes (GLUL, SOX9, AQP4, GJA1, NDRG2, GFAP, ALDH1A1, and ALDH1L1; 13.6%) (Figure 1C–F, Appendix A), endothelial cells (CLDN5, FLT; 2.0%) (Figure 1C–F, Appendix A), excitatory neurons (CAMK2A, NRGN, SATB2, SLC17A7; 21.6%) (Figure 1C–F, Appendix A), inhibitory neurons (GAD1, GAD2; 12.2%) (Figure 1C–F, Appendix A), microglia (C3, CSF1R, CD74, CX3CR1; 6.2%) (Figure 1C–F, Appendix A), oligodendrocytes (MBP, MOG, PLP1, MOBP, and MAG; 36.9%) (Figure 1C–F, Appendix A), CAN oligodendrocyte precursor cells (OLIG1, OLIG2, PCDH15, andPDGFRA; 8.1%) (Figure 1C–F, Appendix A), and unknown cells (0.6%). Among them, oligodendrocyte subsets include c0, c1, c2, c9, and c21; astrocyte subsets include c3, c4, c26, and c28; c8, c10, c11, c12, c14, c16, c19, c22, and c27 are subsets of excitatory neurons; c7, c20, and c29 are subsets of microglia; c5, c15, and c30 are subsets of oligodendrocyte progenitors; c6, c13, c17, c18, and c23 are a subset of inhibitory neurons; c24 are endothelial cells, and c25 are unknown cells.

To determine the major functions of the above eight cell types, we further performed gene set enrichment analysis (GSEA) on DEGs from these eight cell types (Figure 1G). The results showed that the biological functions of astrocytes were mainly concentrated in vascular transport, amino acid transmembrane transport, trans-blood-brain barrier transport, organic anion transmembrane transport activity, amino acid import, carboxylic acid transmembrane transport, organic acid transmembrane transport, L-amino acid transport, L-α-amino acid transmembrane transport, and cellular responses to DNA damage stimuli (Figure 1G). The biological functions of excitatory neurons are mainly enriched in neurotransmitter secretion, synaptic signal release, regulation of synaptic structure or activity, regulation of synaptic organization, regulation of signaling receptor activity, positive regulation of synaptic assembly, anterograde transsynaptic signaling, chemical synaptic transmission, modulation of neurotransmitter receptor activity, and the synaptic vesicle cycle (Figure 1G).

The biological functions of inhibitory neurons are mainly enriched in regulating postsynaptic membrane potential, cation channel activity, ion transmembrane transporter activity, neurotransmitter receptor activity, neurotransmitter secretion, synaptic signal release, synapse assembly, synaptic plasticity, protein localization to cell junctions, and transmembrane transporter activity (Figure 1G). The biological functions of microglia are mainly enriched in immune system processes, leukocyte activation, immune response, immune response regulation, cell activation, positive regulation of immune system processes, regulation of immune system processes, leukocyte-mediated immunity, lymphocyte activation, and positive regulation of the immune response (Figure 1G). The biological functions of oligodendrocytes are mainly enriched in axon wrapping, neuron wrapping, myelination, oligodendrocyte differentiation, oligodendrocyte development, glial cell development, membrane lipid metabolism, plasma cell differentiation, and gliogenesis (Figure 1G). The biological functions of OPC are mainly enriched in chondroitin sulfate proteoglycan biosynthesis, chondroitin sulfate proteoglycan metabolism, embryonic skeletal system development, mucopolysaccharide metabolism, and aminoglycan biosynthesis, as well as the glycosamine glycan biosynthesis, glycosaminoglycan catabolism, and chondroitin sulfate biosynthesis, and chondroitin sulfate metabolism processes (Figure 1G). The biological functions of endothelial cells are mainly enriched in the innate immune response, response to cytokine, regulation of immune effector process, response to interferon-gamma, cellular response to cytokine stimulus, immune system process, defense response to other organism, positive regulation of immune effector process, and cytokine-mediated signaling pathway response to biotic stimulus (Figure 1G). In addition, the biological functions of unknown cells are mainly enriched in actin filament bundle assembly, actin filament bundle organization, Rho protein signal transduction, small GTPase-mediated signal transduction regulation, vascular morphogenesis, cell motility negative regulation, negative regulation of cellular components movement, cell-matrix adhesion, and regulation of stress fiber assembly (Figure 1G).

### 3.3. Integration Analysis of Brain DEGs and Serum DEPs in AD

In order to obtain the differential molecules between the brain and serum of AD patients, we extracted the DEGs from two different brain regions and DEPs from serum of AD individuals (Appendix A), respectively. Then, a differential comparison was carried out between the above brain DEGs and serum DEPs, which uncovered a total of 10 DEGs/DEPs in the endothelial cells, excitatory neurons, microglia, and unknown cells (Figure 2), including NEBL, CHSY3, STMN2, MARCKS, VIM, FGD4, EPB41L2, PLEKHG1, PTPRZ1, and PPP1R14A (Table 2).

### 3.4. GO Functional Enrichment Analysis of DEGs

We performed GO enrichment analysis for 10 DEGs to elucidate their biological functions. GO terms, including biological process (yellow), molecular function (blue), and cellular component (green), are shown in Figure 3. Among these terms, the biological process of 10 DEGs is mainly enriched in actin filament-based process, cytoskeleton organization actin, cytoskeleton organization, supramolecular fiber organization, cardiac muscle thin filament assembly, positive regulation of microtubule depolymerization, negative regulation of neuron projection development, positive regulation of protein localization to the cell cortex, oligodendrocyte progenitor proliferation, and regulation of oligodendrocyte progenitor proliferation. The molecular function is mainly enriched in actin binding, cytoskeletal protein binding, PH domain binding, N—acetylgalactosaminyl—proteoglycan, 3—beta—glucuronosyltransferase activity, glucuronosyl—N—acetylgalactosaminyl—proteoglycan, 4—beta—N—acetylgalactosaminyltransferase activity, keratin filament binding, protein-containing complex binding, actin filament binding, guanyl-nucleotide exchange factor activity, and structural molecule activity. The cell component is mainly enriched in actin filament bundle, perineuronal net, focal adhesion, cell-substrate adherens junction, cell leading edge, cell-substrate junction, germinal vesicle, perisynaptic extracellular matrix, synapse-associated extracellular matrix, and actin cytoskeleton. These results help to elucidate the biological function changes in the AD brain.

### 3.5. KEGG Pathway Analysis of DEGs and PPI Network Construction

KEGG pathway enrichment analysis was conducted on those 10 DEGs to further elucidate their potential function, in which 8 pathways were screened out (Figure 4), including microRNAs in cancer, glycosaminoglycan biosynthesis, chondroitin sulfate/dermatan sulfate, Epithelial cell signaling in helicobacter pylori infection, vascular smooth muscle contraction, Fc gamma R-mediated phagocytosis, Salmonella infection, Epstein–Barr virus infection, and metabolic pathways. The PPI network with obvious interaction relationship for all DEPs included 30 nodes and 119 edges (Figure 5). Then, the top five proteins with the max number of interactors were NEBL, STMN2, MARCKS, VIM, and FGD4.

### 3.6. Expression Changes of Hub DEPs in AD and Control Serum

Because of the high number of interactors and AD-associated functional enrichment of NEBL, EPB41L2, FGD4, and MARCKS in PPI Network and GO analysis, we think they may play a vital role in AD. Therefore, expression levels of NEBL, EPB41L2, FGD4, and MARCKS in serum of AD patients were verified by ELISA. As demonstrated, compared with the control group, the concentrations of NEBL, EPB41L2, and FGD4 were significantly decreased in the AD group (Figure 6A–C), while the concentration of MARCKS protein showed a slight elevation in AD serum, without statistical difference (Figure 6D, *p* = 0.27). Moreover, correlation analysis of age revealed that there are no age-related change trends for these molecules in the control and AD groups (Figure 6E,F). Furthermore, the differential expression analysis of these biomarkers between the female and male populations showed no significant differences in the control and AD groups (Figure 6G,H).

## 4. Discussion

In recent years, the in-depth of brain tissue research and continuous innovation of sequencing methods have improved our understanding of the pathogenesis of AD. The increasing number of pathways and molecules related to AD are being identified, suggesting that the occurrence and development of AD is involved with multiple factors [24,25]. Currently, some studies have demonstrated that amyloid and tau still have their limitations as biomarkers of AD [26]. Moreover, the biomarkers derived from blood or CSF failed to reflect the functional changes of nerve cells. Many limitations indicate that it is necessary to discover new AD biomarkers [27]. Nevertheless, studies on the proteomic changes in the plasma or serum of AD patients are still very rare, due to the technical limitations and complex proteomic changes in serum [28]. The advent of single-cell sequencing technology provides an effective method for studying the cellular heterogeneity in the brain and provides an approach to discover cell-specific biomarkers [29,30,31].

In this study, we innovatively combined the serum proteomic and brain single-cell transcriptomic data to investigate potential biomarkers for the treatment of AD. Relative to those studies, which only centered on single-cell transcriptomics [11] or serum proteomics [19], our study narrowed down the screening range of candidate biomarkers. By integrating snRNA data from brain tissues and proteomics data from serum, we identified 10 significantly differentially expressed molecules, which may be derived from endothelial cells (VIM, FGD4, EPB41L2, PLEKHG1, PTPRZ1, and PPP1R14A), excitatory neurons (STMN2 and MARCKS), microglia (CHSY3), and unknown cells (NEBL). Among these molecules, we note that there are several candidate protein biomarkers that are closely associated with neurodegenerative diseases. VIM proteins, as a type III intermediate filament protein, along with microtubules and actin microfilaments, form the cytoskeleton. Studies have shown that VIM is associated with disease progression and memory decline in AD [32,33,34]. Axonal growth-associated factor STMN2 is necessary for normal axonal outgrowth and regeneration [35,36,37]. The improper splicing of the STMN2 has recently been connected to a variety of neurodegenerative diseases [38,39,40,41]. Although some studies suggest that MARCKS is associated with Aβ production and synaptic plasticity, the present findings are inconsistent [42,43,44,45,46]. Thus, the functions and potential mechanisms of MARCKS in AD need to be further explored. FGD4 is a protein involved in the regulation of the actin cytoskeleton and cell shape. FGD4 is widely involved in the myelination of the vertebrate nervous system, and its deficiency can trigger the demyelination of the peripheral nerves in patients [47,48]. As a protein tyrosine phosphatase, PTPRZ1 is seen as a potential susceptibility gene for schizophrenia [49,50,51]. Study reports suggest that NEBL is associated with small and medium-sized vasculitis in Kawasaki disease, which, combined with GO analysis, suggests a possible association with neuronal projection and recognition [52]. EPB41L2 and PPP1R14A are widely expressed in brain tissues and evolutionarily conserved in rodents and primates, but their specific functions have not been elucidated [53,54]. However, no studies have reported on the roles and functions of PLEKHG1 and CHSY3 in the nervous system. These results suggest that the identifying 10 molecules are extremely promising as biomarkers of AD.

Functional enrichment analysis in this study showed that six proteins (VIM, NEBL, EPB41L2, FGD4, MARCKS, and STMN2) were related to cytoskeleton regulation, and four proteins (NEBL, EPB41L2, FGD4, and MARCKS) were associated with the actin cytoskeleton. As we all know, the pathogenesis of AD is very complex, and there is still no unified conclusion to fully describe the occurrence and development of AD. After decades of unremitting efforts of countless researchers, hypotheses about the pathogenesis of AD have been put forward, including β-amyloid (Aβ) deposition to form senile plaques [55], neurofibrillary tangles (NFT) [56] synaptic loss [57] and so on. Based on this, scientists hope to prevent and treat AD by inhibiting amyloid production, fibrosis, and deposition, but they have not been as effective as expected. Existing drugs can only delay the aggravation of pathological symptoms, but they cannot stop or reverse the disease process. Although Aβ deposition and neurofibrillary tangles are still regarded as the main pathological features of AD, the relationship between these lesions and cognitive impairment in patients remains unclear [58]. Moreover, a recent study has overturned our traditional impression that AD is “extracellular amyloid plaque formation comes first, followed by nerve cell death”, suggesting that “neural cell death comes first, followed by extracellular amyloid plaque appearance” [26]. With the deepening of research, numerous pieces of evidence suggested that synaptic loss is an early event in AD development, and it is associated with cognitive decline [47,48]. Dendritic spines, as tiny protrusions formed on the plasma membrane of neuron dendrites, are the main receiving sites for excitatory synaptic signals, as well as the main site for forming excitatory neuron synapses [59]. Therefore, in the process of learning and memory, synaptic plasticity is closely related to the structure and function of dendritic spines [60]. The actin cytoskeleton is an important structure that constitutes dendritic spines [61], of which the dynamic manifestation is the driving force behind dendritic spines and closely related to synaptic plasticity [62]. The proper functioning of the actin cytoskeleton ensures the integrity of the synaptic structure and function in excitatory neurons [63]. Therefore, the actin cytoskeleton is gradually recognized as one of the important targets of AD [64].

Based on multiple lines of evidence, we believe that the actin cytoskeleton changes in cortical neuron synapses are highly correlated with the development of AD, and can be manifested by some specific markers [65]. Therefore, we further examined the expression changes in serum of four molecules (NEBL, EPB41L2, MARCKS, and FGD4) that are closely related to the actin cytoskeleton. ELISA experiment revealed that the expression of NEBL, EPB41L2, and FGD4 were significantly decreased in AD, while the expression of MARCKS exhibited no significant change in AD. We all know that the development of AD is closely related to the age and gender of the patient [66]. However, our results show that there is no significant correlation of the expression level of these four biomarkers with the age or sex of the patients. This may be attributed to the sample size, disease subtype, and ethnicity. Meanwhile, this also suggested that there is no sex or age bias for these biomarkers in the diagnosis or prognosis of AD. In addition, in the future, researchers should use more advanced technologies for multi-center and large-sample research to discover more valuable biomarkers or therapeutic targets for AD. Interestingly, according to cell origin, we found that FGD4 and EPB41L2 are derived from endothelial cells, and NEBL is derived from unknown cells with similar functions to the endothelial cells. This indicates that the endothelial cells play a considerable role in the synaptic changes caused by AD, which is different from the previous understanding. In the past, it was generally believed that endothelial cells are an important part of the neurovascular unit [67]. In AD, the damage of endothelial cells can affect the normal metabolism of the brain and clearance of amyloid, which, in turn, affects the brain normal function and leads to cognitive deficits of AD patients [68]. However, endothelial cells can affect the plasticity and density of neuronal synapses by secreting some pheromones and using intercellular communication, thereby regulating cognitive function [69]. Thus, our findings suggested that the changes of actin cytoskeleton are correlated with the regulation of synaptic plasticity in AD, suggesting that alterations in these biomarkers may precede the loss of synapses.

Finally, our work revealed that the alterations of the actin cytoskeleton are the most pronounced changes detected in the cerebral cortex and serum of AD patients, which not only provides a strong theoretical basis for the development of AD-specific diagnostic biomarkers, but also contributes to the early diagnosis and development of targeted therapeutic agents in patients with AD in the future. Due to the small sample size, the current results cannot comprehensively explain the relationship of these biomarkers with different age, disease stage, sex, or ethnicity. However, we believe that more samples being included in the future will find more valuable biomarkers.

## 5. Conclusions

Taken together, in this study, we finally identified three protein biomarkers (NEBL, EPB41L2, and FGD4) by integrating data of cortex snRNA-seq and serum proteomic datasets. Hence, the present study might unveil potential novel biomarkers for the diagnosis, treatment, and prognosis of AD.

## Figures and Tables

**Figure 1 brainsci-12-01022-f001:**
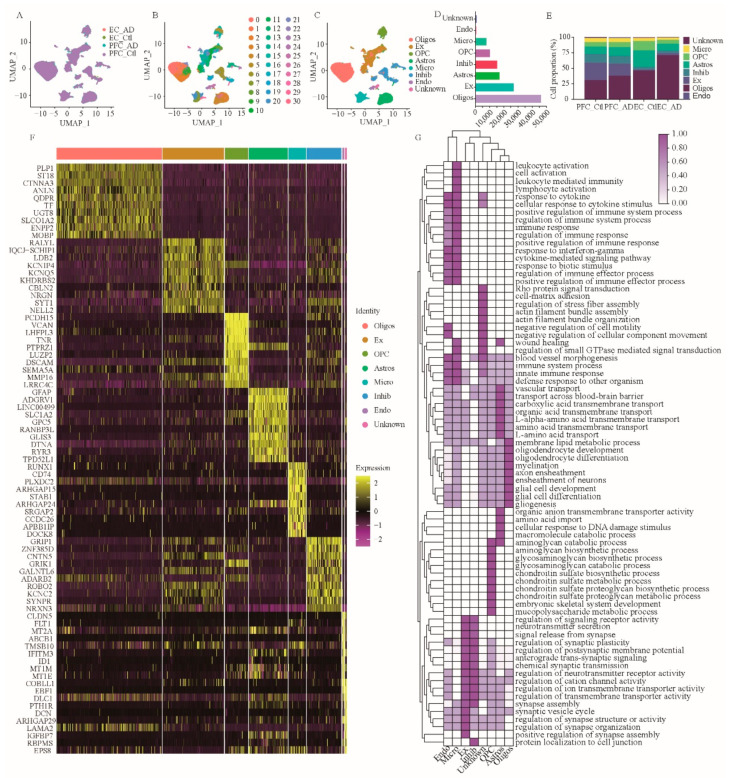
SnRNA-seq analysis of prefrontal and entorhinal cortex of AD patients. (**A**–**C**) Unbiased identification of cell type heterogeneity in human prefrontal and entorhinal cortex. PFC: prefrontal cortex; EC: entorhinal cortex; AD: Alzheimer’s disease; Ctl: Control. A, UMAP plot showing data integration of 3 datasets, 4 groups. B, UMAP plot showing 31 cell clusters obtained after data ensemble cluster analysis. C, UMAP map of seven cell types. (**D**) Number of individual cell types in 124,658 nuclei. (**E**) Cell percentage of the seven cell types in different groups. (**F**) Heatmap of top 10 DEGs in 8 cell types. (**G**) GSEA of cell type-specific DEGs. Oligos: oligodendrocytes; Ex: excitatory neurons; OPC: oligodendrocyte precursor cells; Astros: astrocytes; Micro: microglia; Inhibi: inhibitory neurons; Endo: endothelial cells; GSEA, gene set enrichment analysis.

**Figure 2 brainsci-12-01022-f002:**
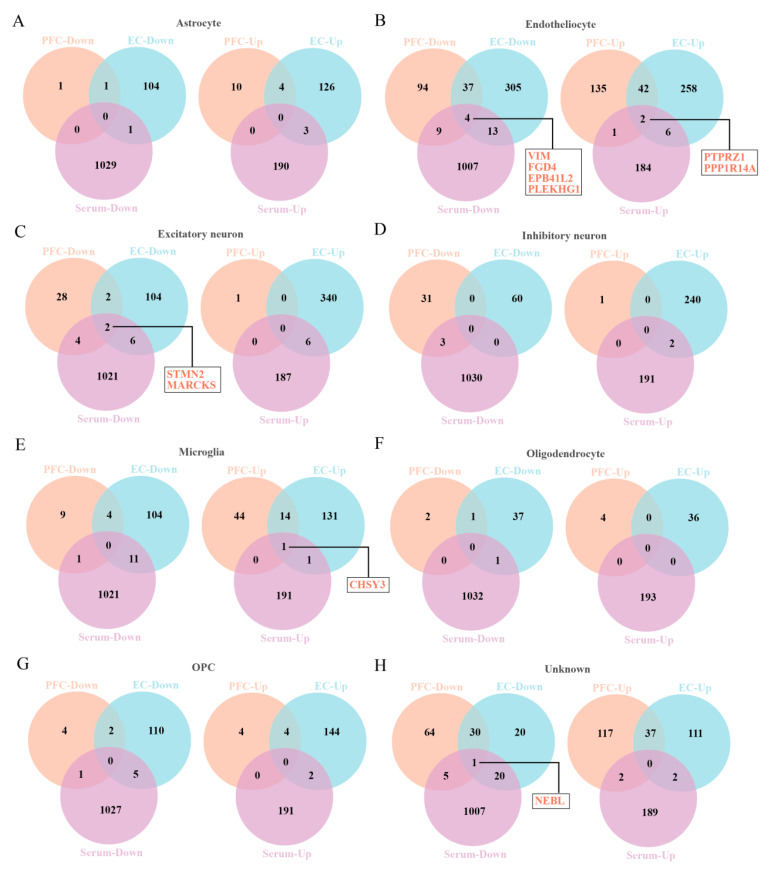
The Venny map overlapped the DEGs of prefrontal cortex and entorhinal cortex and DEPs of serum of AD patients. (**A**–**H**) The DEGs/DEPs for 8 cell types, including astrocytes (**A**), endothelial cells (**B**), excitatory neurons (**C**), inhibitory neurons (**D**), microglia (**E**), oligodendrocytes (**F**), OPC (**G**), unknown cells (**H**).

**Figure 3 brainsci-12-01022-f003:**
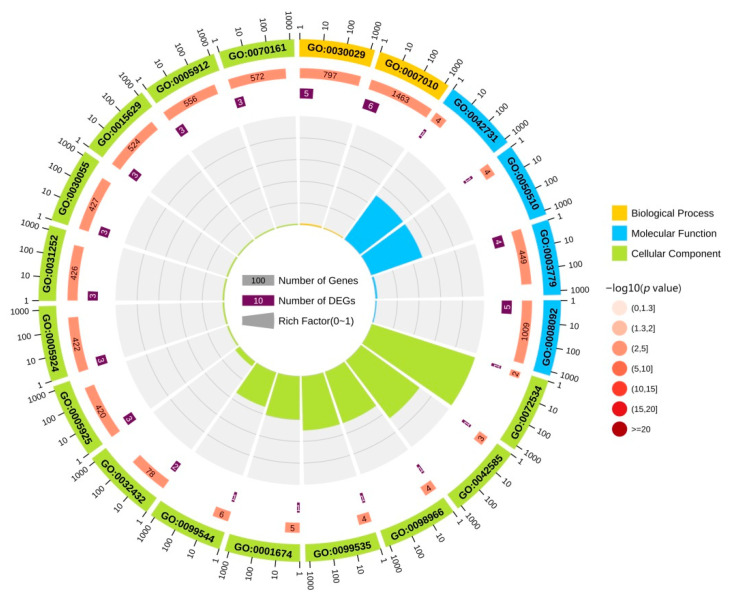
GO functional analysis of DEGs. The yellow area in the figure represents the biological process of DEG. The blue area in the figure represents the molecular function of the DEGs. The green area in the figure represents the cellular component of the DEGs.

**Figure 4 brainsci-12-01022-f004:**
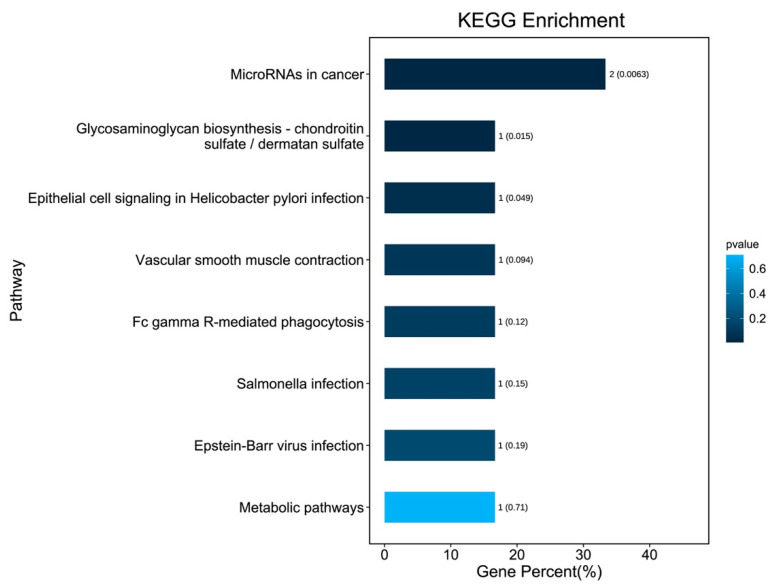
KEGG signaling pathway enrichment analysis for 10 DEGs. The bars indicate the percentage of genes involved in pathway terms; the numbers beside the bars indicate the gene number, and numbers in parentheses represent the *p* value.

**Figure 5 brainsci-12-01022-f005:**
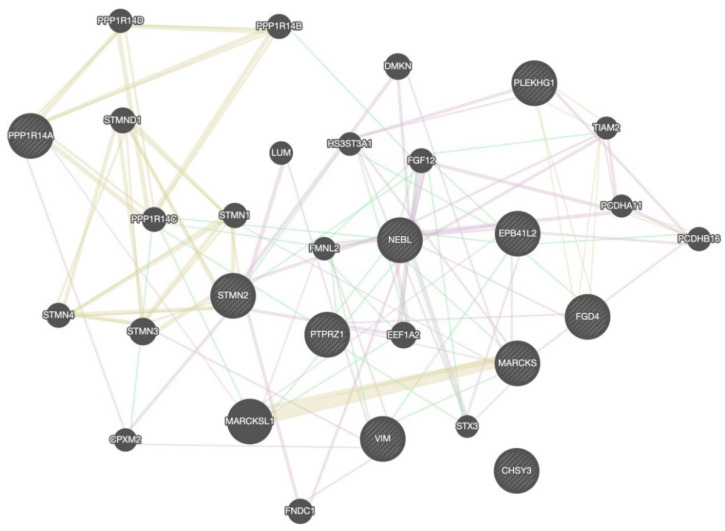
The PPI network analysis for DEGs. The circle in the figure represents the differential proteins. The lines reflect interaction relationship of DEPs. The line color of network edges indicates the type of interaction, and line thickness indicates the strength of data support. The interaction type: co-expression, shared protein domains, and genetic interactions. The set organism: homo sapiens. The size of the circle represents the strength of the relationship between proteins and other proteins.

**Figure 6 brainsci-12-01022-f006:**
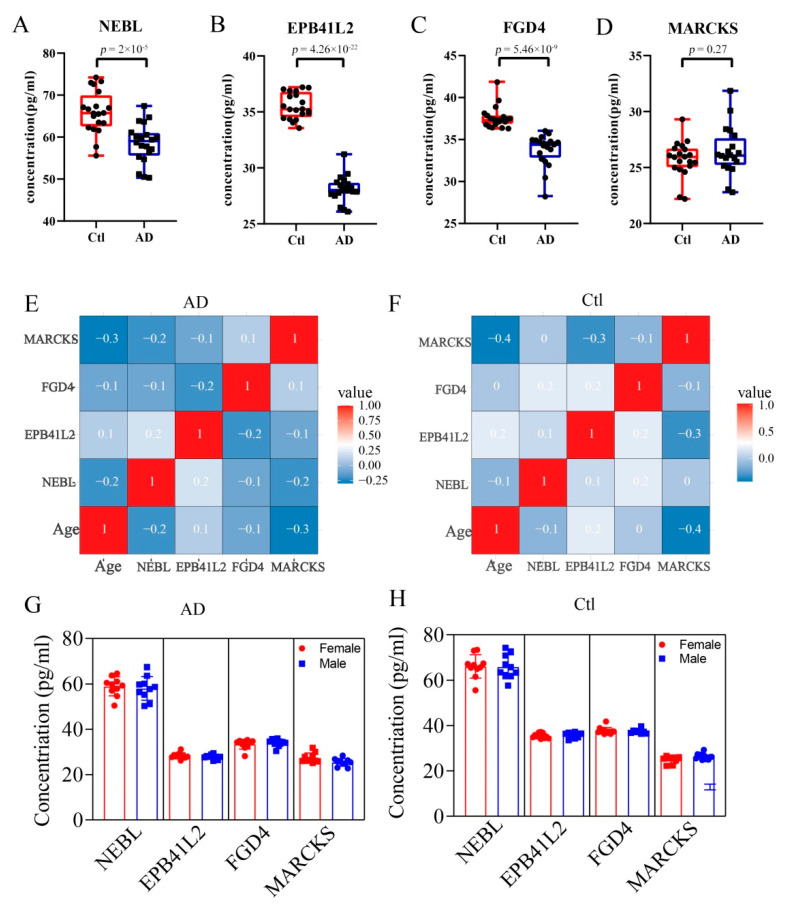
Changes of expression of hub DEPs in the serum. The se-rum concentration of NEBL (**A**), EPB41L2 (**B**), FGD4 (**C**), and MARCKS (**D**) in control and AD groups. (**E**,**F**) Correlation analysis of age with the four molecules in the AD and control groups, respectively. (**G**,**H**) The differential expression analyses of these molecules between female or male populations in the AD and control groups, respectively.

**Table 1 brainsci-12-01022-t001:** Sample information. PFC: prefrontal cortex; EC: entorhinal cortex; Ctl: control.

Tissue Type	Dataset	Total Cases	AD	Ctl	Reference
PFC	GSE141552; GSE157827	18	8	10	[17,18]
EC	GSE138852	12	6	6	[19]
Serum	PXD011482	11	6	5	[20]

**Table 2 brainsci-12-01022-t002:** DEGs/DEPs in AD. Endo: endothelial cells; Ex: excitatory neurons. The “down” represents a down-regulation of this gene expression in AD individuals compared to controls. The “up” represents a up-regulation of this gene expression in AD individuals compared to controls.

Source	Gene Name	Protein Name	Gene Expression in AD
Endo	VIM	Vimentin	Down
Endo	FGD4	FYVE, RhoGEF, and PH domain containing 4	Down
Endo	EPB41L2	Erythrocyte membrane protein band 4.1 like 2	Down
Endo	PLEKHG1	Pleckstrin homology and RhoGEF domain containing G1	Down
Endo	PTPRZ1	Protein tyrosine phosphatase receptor type Z1	Up
Endo	PPP1R14A	Protein phosphatase 1 regulatory inhibitor subunit 14A	Up
Ex	STMN2	Stathmin 2	Down
Ex	MARCKS	Myristoylated alanine rich protein kinase C substrate	Down
Microglia	CHSY3	Chondroitin sulfate synthase 3	Up
Unknown	NEBL	Nebulette	Down

## Data Availability

Informed consent was obtained for all data used in the article. All data generated or analyzed during this study are included in the submitted article (and its Appendix A).

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
