# Peer review of "Integrated Analysis of Cortex Single-Cell Transcriptome and Serum Proteome Reveals the Novel Biomarkers in Alzheimer’s Disease"

_brainsci, 2022, doi:10.3390/brainsci12081022_

Round 1

Reviewer 1 Report

I appreciated that the authors have addressed the topic of the search for biomarkers for the study and diagnosis of AD. The question to be answered and the hypotheses are easily identifiable. However, this article needs a big improvement from many points of view. In the first place, it is difficult to read due to the number of typographical, spelling, and syntactic errors. Sometimes it is impossible to understand what the authors intend to communicate. A review by a native English speaker with knowledge of scientific language is required.

On the other hand, the material and methods part must be rewritten and better organized. In my opinion, the subjects, or patients, as well as the groups that have been used for the study, must appear first, followed by the ethical protocols that have passed to carry out the study. An unification of abbreviations is necessary (for example, in the case of controls that appear as C, Ctl, ... there is an international convention for the abbreviations that must be used).

Reviewer 2 Report

In the manuscript by Yu and coauthors it was investigated which genes differ in expression in a single-cell transcriptome dataset consisting of tissues from controls and AD patients. The authors also performed a protein analysis using ELISA of some of the proteins that were identified in the RNA seq analysis. Although I support the idea of using available datasets for analysis, there are still many improvements to the manuscript that has to be done before the study is convincing. 

The whole manuscript is in need of a careful edition/revision of the text to remove typos and unclear sentences, which are preventing the reader from fully understanding the findings.  

Specific comments:

Abstract:

Line 19: authors claim that proteomic analyses in blood "are very unconvincing". I am not convinced by this sentence and the authors need to explain the arguments supporting this statement, considering the number of successful published studies of AD biomarkers in blood. 

Introduction:

line 36: "(AD), referred to as AD", only one of the explanations is needed. 

"AD" and "Alzheimer's disease" are not consequently used throughout the text. Please use one way of writing the disease. 

line 53: "screening out" can be replaced with "screening for"

line 54: rephrase the sentence ending with "has a very positive sign for patients". I suspect "sign" is not the appropriate word here. "Implication" could be an alternative. 

line 56: "is to" is not a good expression here. Please rephrase. 

Line 74, 75: "datasets from" rather than "datasets in"

More background to the datasets would be appropriate in the introduction for the readers that have not used these datasets before. Also information on how the datasets were chosen.

Material and Methods:

Please better describe the dataset used under 2.1 (see also above). For example, the info in 2.2.3. could be explained already in 2.1.

When reading about GSE141552 in GEO Accession viewer the PFC data comes from 3 alcoholic persons and 4 controls.  Where the brain data from alcoholics included in the analysis? Moreover, in GSE157827 data came from 11 AD patients and 9 controls. In Table 1 of this manuscript it is stated that data from the two datasets came from 8 AD and 10 controls, although from the two databases a total of 11 AD and 13 controls were available. Which two AD persons were excluded and  why were controls selected from two datasets (but 3 controls where then omitted)? This needs to be explained. 

In dataset GSE138852, there is a total of 8 samples from two patients each. In Table 1 is stated that the number of participants were 6 of each diagnosis (control and AD). Please explain the discrepancy. 

Explain why the specific tissues and brain regions were chosen from available datasets.

In general, the Material and Method section can be improved by including more details regarding how the analyses were performed.  

Line 87: "loaded into the R" could probably be rephrased to read better

Line 118: Explain NC here.  

2.5 ELISA validation: More information is needed to describe the methods used in this paragraph. Moreover, order/catalog numbers of the ELISA kits need to be reported. 

What was the sex of the serum donors, and did the ELISA data differ between the two sexes?

Result:

Table 1: Please avoid abbreviations in the table or explain these in a legend. Moreover, add an "s" at the end of Total case (Total cases)

Figure 1: Lettering for individual panels needs to be larger. Also text in the panels would benefit from being written with larger font

Table 2: Explain that the comparisons were between AD patients and controls in the Table title. Instead of Endo and Ex, please write out the whole words or include a legend with abbreviations. For the column for up- and down-regulation, make sure the reader understands this is AD in relation to controls. 

Line 261: Here Figure 4 is referred to as a "bubble diagram" but the graph shows rectangles (bars) so the term could be wrongly used. 

Figure 4: Please explain the numbers and numbers in parentheses to the right of each bar in this diagram.

Figure 5: Note that there are two figures named 5

3.6 ELISA validation results: How/why where the four genes selected for ELISA analysis? Why were not all the genes investigated from Table 2? Explain how the tested proteins were chosen. 

Discussion: 

The discussion is mostly focusing on the ELISA data, while it would be useful to also include discussions and comparisons of the findings from the gene expression data as well. 

Moreover I would like to see a discussion of the implications of single cell transcriptomics as a methods to identify possible biomarkers. 

A large portion of the discussion is focusing on the MARCKS gene/protein despite the fact that this protein was not differently expressed in AD serum compared to controls while less is written regarding the other investigated serum proteins. 

MARCKS is spelled and misspelled in many ways in this section, please perform a throughout proofreading of the manuscript. 

I find the sentence "We demonstrate the role of vascular endothelial cells..." (lines 394 to 395) needs to be changed as this is overstating the authors' findings in this manuscript. The same goes for the sentence on line 402 to 404: In this study, we demonstrate that the kinetic alteration..." which is to exaggerate the findings. 

Line 399: Sentence starting with "Find out if ..." seems incomplete. 

The manuscript would benefit from a clearer description of future research as well, as sections describing the strengths and weaknesses of the study.

It would also be good to see the findings in perspectives of other studies of the same RNA seq material? How did other results from other studies with similar methods/data differ/match to the present study? For example, how did the findings differ/match the paper of Grubman et al who used the same material?

Please explain how your study has contributed to further findings in relation to the already published papers of using these datasets. 

The manuscript would also benefit from the summarizing the data in a short conclusions section. 

Reviewer 3 Report

Authors performed extensive proteomic biomarkers and correlated with cortex single-cell transcriptome to their application in AD.

English grammar needs much improvements in verb tenses and punctuations. Better transition between paragraphs can focus the results.

Authors need to discuss the time course expressions of those biomarkers from young to disease status. It would be interesting to measure and correlate with other plasma biomarkers.

Authors also need to perform the literature searches of those biomarkers in human and their molecular significances.

Reviewer 4 Report

Authors should correct the abbreviations. Differentially expressed genes (DEG) is abbreviated as DGE in many places.

Also, check the whole manuscript thoroughly to avoid such mistakes.
